# Prevalence of Osteoporosis Assessed by DXA and/or CT in Severe Obese Patients

**DOI:** 10.3390/jcm11206114

**Published:** 2022-10-17

**Authors:** Marion Halin, Edem Allado, Eliane Albuisson, Laurent Brunaud, Isabelle Chary-Valckenaere, Damien Loeuille, Didier Quilliot, Marine Fauny

**Affiliations:** 1Department of Rheumatology, University Hospital, F-54000 Nancy, France; 2University Center of Sports Medicine and Adapted Physical Activity, CHRU-Nancy, F-54000 Nancy, France; 3DevAH, Université de Lorraine, F-54000 Nancy, France; 4Unité de Méthodologie, Data Management et Statistiques (UMDS), Département MPI, DRCI, CHRU-Nancy, F-54000 Nancy, France; 5IECL, CNRS, Université de Lorraine, F-54000 Nancy, France; 6Unité Multidisciplinaire de la Chirurgie de L’obésité (UMCO), University Hospital, F-54000 Nancy, France; 7Inserm UMRS 1256 N-GERE (Nutrition-Genetics-Environmental Risks), Faculty of Medicine, University de Lorraine, F-54000 Nancy, France; 8Department of Digestive, Hepato-Biliary and Endocrine Surgery, University Hospital, F-54000 Nancy, France; 9Ingénierie Moléculaire et Physiopathologie Articulaire (IMoPA), UMR 7365 CNRS—University of Lorraine, F-54000 Nancy, France; 10Department of Endocrinology Diabetology and Nutrition, University Hospital, F-54000 Nancy, France; 11Department of Rheumatology, Saint Charles Hospital, F-54200 Toul, France

**Keywords:** obesity, bone mineral density (BMD), scanographic bone attenuation coefficient (SBAC-L1), tomodensitometry/CT, dual-energy X-ray absorptiometry (DXA)

## Abstract

The primary objective was to evaluate bone fragility prevalence on dual X-ray absorptiometry (DXA) and computed tomography (CT) in patients with severe obesity. The secondary objective was to evaluate the risk factors for bone fragility. This monocentric study was conducted in patients with grade 2 and 3 obesity. Bone mineral density (BMD) and T-score were studied on DXA, and the scanographic bone attenuation coefficient of L1 (SBAC-L1) was measured on CT. Among the 1386 patients included, 1013 had undergone both DXA and CT within less than 2 years. The mean age was 48.4 (±11.4) years, 77.6% were women, and the mean BMI was 45.6 (±6.7) kg/m². Eight patients (0.8%) had osteoporosis in at least one site. The mean SBAC-L1 was 192.3 (±52.4) HU; 163 patients (16.1%) were under the threshold of 145 HU. Older age (OR[CI95] = 1.1 [1.08–1.16]), lower BMD on the femoral neck and spine (OR[CI95] = 0.04[0.005–0.33] and OR[CI95] = 0.001[0.0001–0.008], respectively), and higher lean mass (OR[CI95] = 1.1 [1.03–1.13]) were significantly associated with an SBAC-L1 ≤ 145 HU in multivariate analysis. Approximately 16% of patients with severe obesity were under the SBAC-L1 threshold, while less than 1% were classified as osteoporotic on DXA.

## 1. Introduction

Obesity is defined as an abnormal or excessive accumulation of fat [1]. Obesity is often associated with other comorbidities, such as diabetes, dyslipidemia, hypertension, or obstructive sleep apnea [2]. Body mass index (BMI) is used to estimate overweight in adults regardless of age or sex: mild obesity is defined by a BMI from 30 to 34.9 kg/m², severe obesity is defined by a BMI between 35 and 39.9 kg/m² (Grade 2), and morbid obesity is defined by a BMI ≥ 40 kg/m² (Grade 3) [3]. Obesity has become a chronic public health problem that is constantly rising worldwide [4].

Osteoporosis is defined by bone fragility due to low bone mineral density and/or microarchitecture alterations leading to a higher risk of fracture and by a value for bone mineral density or content (BMD/BMC), 2.5 SD or more below the young adult mean [5,6]. Osteoporosis is a public health problem due to the consequences of fractures in terms of residual physical disability, loss of autonomy, institutionalization, and, above all, higher mortality for major fractures [7]. Measurement of bone mineral density (BMD) with dual X-ray absorptiometry (DXA) remains the gold standard [6] for the diagnosis of osteoporosis.

In patients with obesity, DXA has some limitations, such as reduced photon penetration through soft tissues [8]. Thus, BMD measurements by DXA increase with BMI [9]. In the literature, DXA evaluation in patients with obesity did not show more frequent osteoporosis than in subjects without obesity [10], but some studies highlight that people with obesity had more fractures, especially limb fractures [11,12,13], with higher BMD than patients without obesity [10].

Computed tomography (CT) is often performed in patients with obesity before bariatric surgery to detect malformation and prevent operative complications. CT allows both a qualitative bone evaluation with the study of vertebral fractures (VF) and a quantitative bone evaluation with the scanographic bone attenuation coefficient of the first lumbar vertebra (SBAC-L1) measure [14,15,16,17,18,19,20]. An SBAC-L1 ≤ 145 Hounsfield Units (HU) is more sensitive than a T-score ≤ −2.5 SD in identifying vertebral fracture risk [14]. The threshold at 145 HU allows the best compromise between sensitivity and specificity to screen patients at risk of vertebral fracture [15,16,17,18]. Moreover, this technique is less prone to artifacts, avoiding cortical bone, osteophytes, vascular calcifications, and fat mass.

No study has analyzed both the BMD data and the bone attenuation coefficient of the first lumbar vertebra scan in a population of severely obese subjects.

Based on these data, we conducted an observational study whose primary objective was to assess the prevalence of bone fragility on DXA (defined by a T-score ≤ −2.5 SD) and on CT (defined by SBAC-L1 threshold ≤ 145 HU) in patients with severe or morbid obesity (Grade 2 and 3) before bariatric surgery. The secondary objectives were to study the prevalence of vertebral fractures on CT and to assess the risk factors for bone fragility in patients with severe or morbid obesity (Grades 2 and 3).

## 2. Materials and Methods

### 2.1. Population

This descriptive, observational, monocentric, and retrospective study was conducted on patients with grade 2 and 3 obesity who were followed in the Lorraine obesity center, underwent bariatric surgery between January 2014 and December 2018 and underwent CT and/or DXA before surgery. All patients included met the criteria for bariatric surgery: BMI > 40 kg/m² or BMI > 35 kg/m² associated with at least one comorbidity (notably hypertension, obstructive sleep apnea syndrome, type 2 diabetes, severe joint disease, etc.). All the exams had to be performed at Nancy University Hospital during the routine preoperative screening. The exclusion criteria were gastric banding surgery and ring ablation.

Demographic and anthropometric data were collected, such as age, sex, smoking and alcohol status, height, and weight (at the time of DXA and CT), to calculate body mass index, diabetes and cardiovascular risk factors (dyslipidemia, hypertension or NASH), and available osteoporosis risk factors (sex, age, smoking and alcohol consumption, vitamin D deficiency defined by a level below 30 ng/mL).

### 2.2. DXA Evaluation

DXA was performed on a Lunar Prodigy densitometer (Advance PA +301010, Encore, version 14.10.022; Madison, WI 53718, USA). The BMD and the T-score were assessed for each patient at the lumbar spine, femoral neck and total hip. The diagnosis of osteoporosis and bone fragility was retained for a T-score ≤ −2.5 SD at any measured location. Osteopenia was defined by −2.5 SD < T-score ≤ −1 SD [5]. The BMD measurement was coupled with an analysis of the distribution of fat and lean mass on the same DXA.

### 2.3. CT Evaluation

Any available preoperative CT was used if it included the first lumbar vertebra. The preoperative CT closest to the date of the surgery was selected. The CT scans were analyzed on Synapse Mobility Web, V.6.0, 2016, FUJIFILM Medical Systems U.S.A., Inc.; Miami, FL 33186, USA.

Axial acquisitions allowed SBAC-L1 measurement in the bone window on the pedicle slice. The largest elliptical region of interest (ROI) was drawn in trabecular bone, avoiding cortical bone, and provided the average bone mineral density in HU. The threshold of 145 HU defined by Pickhardt et al. was used as a threshold to classify patients with bone fragility [14,18]. This threshold allowed the best compromise between sensitivity and specificity in the general population [14].

This measure was performed by a single investigator (who was blinded to the DXA results and clinical data) because excellent intra- and interreader reliabilities have been previously described (ICC > 0.9) [17]. In cases of fracture or discovertebral damage with osteosclerosis of the vertebral endplate of L1, SBAC was performed on L2 or on the first adjacent vertebra without VF.

### 2.4. Vertebral Facture

Sagittal reconstructions allowed manual VF analysis, according to an adaptation of the Genant classification, which is usually used on spine radiographs [21]. The grade of the VF was determined by the most severe lesion observed on sagittal sections.

### 2.5. Ethics Approval

All the data used were obtained from the medical records. No patient examination was performed to meet the inclusion criteria. This study was registered to the Information Technology and Freedoms Commission for the University Hospital of Nancy and was designed in accordance with the general ethical principles outlined in the Declaration of Helsinki. The protocol of this study was approved by the Information Technology and Freedoms Commission for the University Hospital of Nancy. All patients gave their consent before surgery for the use of their medical data for future research. The study was registered with clinical trial identifier N° NCT04174495.

### 2.6. Statistical Analysis

Both descriptive and comparative analyses were conducted by accounting for the nature and distribution of the variables (normality assessed by the Kolmogorov–Smirnov test). Qualitative variables were described with frequencies and percentages; quantitative variables were evaluated with the mean ± SD for variables with a normal distribution and the median (interquartile range) for variables with a nonnormal distribution.

Student’s *t* test was used for age, and the Mann–Whitney U test was used for the other variables. For qualitative variables, the chi-square test and/or Fisher’s exact calculation was used. Logistic regression was performed to test variables significantly associated with the binary outcome SBAC-L1 ≤ 145 HU and of clinical interest (we included only significant variables from the univariate analysis, variables with missing data were not included (like VF, smoking status, alcohol et vitamin D). Osteopenia, osteoporosis and T-score were not included because they are associated with BMD. Ratio variables were also excluded). Significant results (univariate and multivariate analysis) are presented with the odds ratio (OR) and its 95% confidence interval (CI 95%). To study the correlation between CT and DXA results, the Pierson coefficient was used. The significance level was set at 0.05 for the entire study. IBM SPSS Statistics v23 was used for the data analysis.

## 3. Results

### 3.1. Population

Of 2146 patients with grade 2 and 3 obesity who underwent bariatric surgery at Nancy University Hospital between January 2014 and December 2018, 1386 patients were included, as they underwent at least a CT and/or a DXA before surgery during routine preoperative screening. Among them, 1253 patients underwent DXA, 1179 underwent CT before surgery, and 1046 patients underwent both DXA and CT before surgery; 1013 patients underwent these two exams within less than 2 years (Figure 1).

In our population of 1013 patients, the mean age was 49.0 (±18.0) years, with a large majority of women (76.6%). The mean BMI was 44.6 (±8.3) kg/m^2^. Most patients (90.6%) underwent gastric bypass (GBP) (n = 918), nearly 10% underwent sleeve gastrectomy (n = 94, 9.3%), and only one patient (0.1%) had a single anastomosis (SADI). In total, 842 patients (91.3%) presented vitamin D deficiency. Approximately 40% of patients were current smokers.

The details of the demographic and anthropometric characteristics of the patients are presented in Table 1.

### 3.2. Patients with Both CT and DXA Evaluation within 2 Years (n = 1013)

The mean SBAC-L1 was 194.0 (±69.0) HU; 163 patients (16.1%) had an SBAC-L1 ≤ 145 HU. VF was observed in 22 patients (2.2%). Only 8 patients (0.8%) presented osteoporosis at least on-site on DXA, and 119 patients (11.7%) an osteopenia (Table 1).

Of these patients with VF, 12 (44.4%) presented an SBAC-L1 ≤ 145 HU, whereas only one of these patients (3.7%) had osteoporosis on DXA at least one site (on the spine, none on the femoral neck). Osteopenia at one site was observed in 9 patients (33.3%) (Table 2, Figure 2). The average time between the exams was 275.4 (±174.4) days. Concerning voltage, most of the CTs are performed with a 120 kV voltage (min 100 kV—max 140 kV, adapted to fat mass quantity). The mean slice thickness was 1.80 (±0.91) mm. Voxel size was not available.

For these patients, an SBAC-L1 ≤ 145 HU was significantly associated with higher age (OR[CI95] = 1.1 [1.10–1.14]), diabetes (OR[CI95] = 1.9 [1.35–2.68]) and cardiovascular risk (OR[CI95] = 2.7 [1.84–3.96]) in univariate analysis (Table 2). Tabaco, alcohol and vitamin D deficiency were not significantly associated with SBAC-L1 status (*p* = 0.11, 0.07, and 0.5, respectively). Female and patients with high BMI were less likely of having SBAC-L1 ≤ 145HU (OR[CI95] = 0.4 [0.27–0.55] and OR[CI95] = 0.9 [0.94–0.99], respectively). An SBAC-L1≤ 145 HU was also associated with lower BMD or T-score and the presence of osteopenia at each site (femoral neck, hip, and spine). The presence of osteopenia on the femoral neck and hip was associated with a higher risk of having an SBAC-L1 ≤ 145 HU (OR[CI95] = 11.1 [6.45–18.97] and OR[CI95] = 5.6 [2.51–14.48], respectively). Osteoporosis of the spine was associated with an OR[CI95] = 9 [2.12–38.02]. Higher lean mass was significantly associated with an SBAC-L1 > 145 HU (OR[CI95] = 1.02 [1.01–1.04]).

For the multivariate analysis (Table 2), all significant variables in the univariate analysis were included. The risk of developing SBAC-L1 ≤ 145 HU increased when age increased (OR[CI95] = 1.1 [1.08–1.16]), when femoral neck and spinal BMD decreased (OR[CI95] = 0.04 [0.005–0.33] and OR[CI95] = 0.001 [0.0001–0.008], respectively), and when lean body mass increased (OR[CI95] = 1.1 [1.03–1.13]). Higher BMI was a protective factor for SBAC-L1 ≤ 145 HU, with an OR of 0.9 [0.82–0.92]. The presence of diabetes or cardiovascular risk factors was not significantly associated with an SBAC-L1 under the threshold of 145 HU in multivariate analysis.

### 3.3. Discordance Analysis between CT and DXA

Three patients were considered osteoporotic at the spine on DXA but had an SBAC-L1 > 145 HU (151, 192, and 202 HU); these patients were two women aged 60 years and one man aged 30 years; their mean BMI was 45 (±6.4) kg/m². All were diabetic, two were smokers, and two had vitamin D deficiency. Two of them also had a T-score ≤ −2.5 SD at the femoral neck. None of them had a vertebral fracture on CT.

In contrast, 149 patients had an SBAC-L1 ≤ 145 HU but had no osteoporosis (at any site) on DXA, and 92 patients had a normal BMD (no osteoporosis and no osteopenia) at any site. There were 43 men and 49 women. Two (2.2%) of them had VF. The mean age was 57 (±7.8) years, the mean BMI was 44.2 (±6.3) kg/m², 44.6% (n = 41) were smokers, 81.5% (n = 75) had vitamin D deficiency, and 41.3% (n = 38) were diabetic.

The correlation coefficient was 0.48 (CI95 = 0.43–0.53) between femoral neck T-score and SBAC-L1 and 0.43 (CI95 = 0.36–0.49) between femoral neck BMD and SBAC-L1; 0.46 (CI95 = 0.40–0.51) between spine T-score and SBAC-L1 and 0.42 (CI95 = 0.37–0.48) between spine BMD and SBAC-L1 (Pearson coefficient was used for BMD and Spearman’s rho for T-score, according to the variables distribution). So, the correlation was poor but positive.

## 4. Discussion

This study evaluated the prevalence of bone fragility on DXA (T-score ≤ −2.5 SD) and CT (SBAC-L1 ≤ 145 HU) and the prevalence of VF on CT in patients with severe and morbid obesity before bariatric surgery. To our knowledge, this is the first study to investigate bone fragility on CT in a population with grade 2 and 3 obesity.

In our study, among the 1013 patients who underwent both DXA and CT within two years, 0.8% were considered osteoporotic, and 11.7% had a low bone mass on DXA. This percentage was lower than those observed in the literature, with a prevalence varying from 1.8% to 8% for osteoporosis [22,23] and from 29% to 51.6% for low bone mass [10,22,24]. Using the SBAC-L1, 17% of the patients had an SBAC-L1 under the threshold of 145 HU; these results suggest an underestimation of the bone risk in patients with obesity through DXA. CT seems to be more sensitive in this population for bone fragility screening using the SBAC-L1. In view of the discordances between CT and DXA, we did not find any relation between T-scores and SBAC-L1. This lack of relationship could be explained by the difference in terms of measure between the 2 exams. CT excluded cortical bone and is less influenced by the body fat, and the values obtained are more representative of the real bone density than those obtained by DXA. Moreover, one advantage of CT compared to DXA is its ability to accurately identify unsuspected osteoporotic vertebral fractures, which clearly diagnoses osteoporosis independent of the patient’s DXA T-score. DXA remains the gold standard examination for osteoporosis screening, but it may not be the most reliable examination in patients with obesity [24,25] because BMD measurement may introduce errors due to increased body fat [25]. The SBAC-L1 measure is an alternative to specifically exploring trabecular bone while avoiding the cortical bone. The threshold of 145 HU was used because it allowed the best compromise between sensitivity and specificity in a general population [14], but no study has been previously conducted in a population with obesity. This population may have been a bit young to expect a detectable level of bone degradation that would be classified by DXA as osteoporosis or osteopenia; it makes a stronger case for the use of CT evaluation for early bone fragility and fracture risk. However, CT is not feasible just to explore bone fragility unless there is vertebral fracture suspicion. This examination is expansive and exposes the patient to too much radiation compared to DXA. However, CTs are very frequently performed for other indications, so we can use them opportunistically to screen patients at risk of bone fragility. In the future, a new low-dose spine CT could offer a new approach to detecting vertebral fractures and measuring SBAC-L1. 

For VF evaluation, 27 patients (2.3%) among 1179 who underwent CT had at least one VF on CT for a total of 41 VFs. Among them, 40.9% had low bone mass, 4.5% were osteoporotic on DXA and 44.4% had an SBAC-L1 under the threshold of 145 HU; these results are in accordance with the literature, with a percentage of VF ranging from 0.6 to 4.2% [26].

In multivariate analysis, all significant variables in univariate analysis were included. sex and diabetes were not associated with an SBAC-L1 ≤ 145 HU. Similar results are found in the literature [2,19]. Diabetes appeared to be a confounding factor that may be related to age or BMI [24]. An SBAC-L1 ≤ 145 HU was statistically significantly associated with older age (OR(CI95) = 1.1 [1.08–1.16]) and lower BMI (OR(CI95) = 0.9 [0.82–0.92]) and with higher lean mass (OR(CI95) = 1.1 [1.03–0.1.13]). In the literature, BMI is sometimes considered a risk factor for osteoporotic fractures [23,24,27] and is sometimes considered protective [11]. High fat mass was previously associated with a higher prevalence of bone fragility on DXA [23,24,27]. This may be explained by reduced mobility, higher fall frequency, inflammatory stress, inadequate levels of leptin, adiponectin, and ghrelin, hypoestrogenism and hyperinsulinism, storage of vitamin D in adipose tissue, and elevated levels of parathyroid hormone [28,29]. An SBAC-L1 ≤ 145 HU was also associated with a lower BMD on the spine and femoral neck. A high fat mass has been associated with many factors that could affect bone health and fragility, but the results did not show a strong influence of BMI on the SBAC-L1.

The presence of VF was significantly associated with an SBAC-L1 ≤ 145 HU in univariate analysis but was not studied in multivariate analysis because of the small number of events in our population. Unfortunately, any studies could be performed according to the presence or absence of VF due to the small number of patients with VF. Furthermore, any comparison between patients with or without osteoporosis on DXA could be performed due to the low number of patients with osteoporosis (only 3 patients).

The correlation between DXA and CT was positive but poor. This lack of relationship could be explained by the difference in term of measure, between the 2 exams. CT excluded cortical bone and is less influenced by the body fat mass, and is more representative of the real bone mass, than DXA. Moreover, one advantage of CT compared to DXA is its ability to accurately identify unsuspected osteoporotic vertebral fractures, which clearly diagnoses osteoporosis independent of the patient’s DXA T-score.

Our study had some limitations. The first limitation was the retrospective nature of the study, which led to a lack of available data in the clinical records. For example, we did not have all the data concerning risk factors for osteoporosis, such as menopausal status, specific treatment for osteoporosis or previous fracture, which were not always mentioned in the medical records. Fractures were collected by blinded CT reading, so we had no clinical data on the traumatic or nontraumatic context. Our morphological examination focused on the available vertebrae from T10 to S1 in 95% of cases (877 abdominopelvic CT); therefore, evaluation of the thoracic spine was unavailable. The risk of underestimating the prevalence of VF must be taken into account; however, we know that a large proportion of osteoporotic VF occur at the thoracolumbar junction [20]. Data on weight changes over time between DXA and CT were not available. DXA and CT allowed only a quantitative bone evaluation. No qualitative study of the bone could be performed with these exams.

The strengths of this work are the large number of included patients and the ability to conduct a bone fragility evaluation without incurring additional costs, radiation, or time.

## 5. Conclusions

We showed for the first time in a large population of patients with grade 2 and 3 obesity that 17% of the population had an SBAC-L1 under the threshold of 145 HU, while less than 1% were classified as osteoporotic on DXA. Risk factors for SBAC-L1 ≤ 145 HU were older age, lower BMI, low BMD at the femoral neck or spine and high lean mass; these results suggest an underestimated risk of bone fragility in this population with DXA.

SBAC-L1 measurement may be a complementary alternative to exploring bone fragility in patients with obesity without artifacts due to fat mass. Further analysis could be performed on CT to assess bone fragility (defined by a SBAC-L1 under 145 HU) in patients with obesity after bariatric surgery.

In the future, new low dose spine CT could offer a new approach to detecting vertebral fracture and measuring SBAC-L1. CTs are very frequently performed for other indications, so we can use them opportunistically to screen patients at risk of bone fragility.

## Figures and Tables

**Figure 1 jcm-11-06114-f001:**
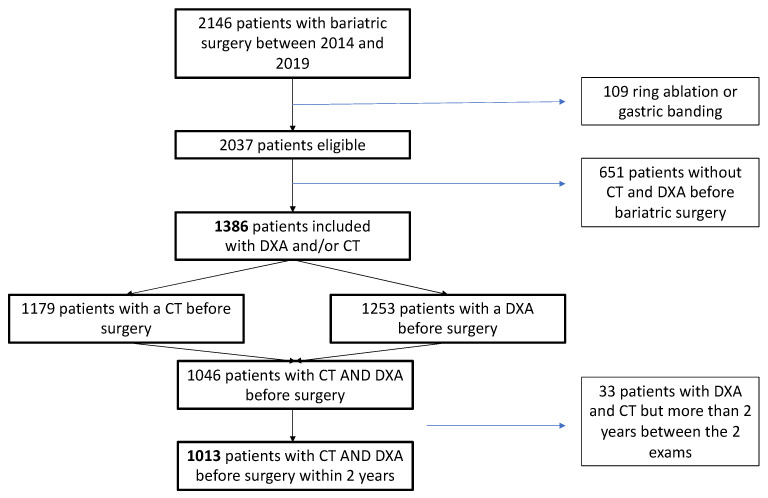
Flow chart. CT: computed tomography; DXA: dual-energy X-ray absorptiometry.

**Figure 2 jcm-11-06114-f002:**
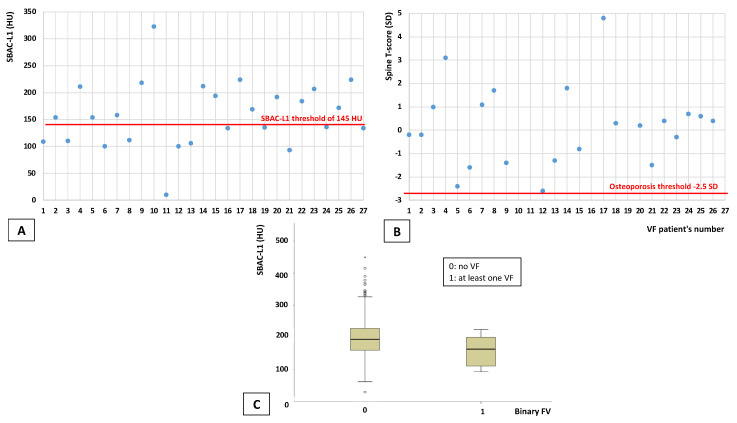
Distribution of patients with VF according to the bone fragility assessment method (SBAC-L1 ≤ 145 HU for A and C and T-score ≤ −2.5 SD for B). DXA: dual-energy X-ray absorptiometry; CT: computed tomography; SBAC-L1: Scanographic bone attenuation coefficient. (**A**): Distribution of patients with VF according to the SBAC-L1 threshold (CT evaluation). (**B**): Distribution of patients with VF according to the femoral neck T-score (DXA evaluation). (**C**): Distribution of SBAC-L1 for patients with or without VF.

**Table 1 jcm-11-06114-t001:** Patient characteristics.

	DXA and CT
**Demographical Data**	**n = 1013**
Age (years)	49.0 (±18.0)
Sex (women)	776 (76.6%)
BMI (kg/m²)	44.6 (±8.3)
Diabetes	316 (31.2%)
Cardiovascular risk factor	584 (57.7%)
Tabaco Smokers Never smokers	412 (42%) 569 (58%)
Alcohol	42 (4.3%)
Vitamin D deficiency	842 (91.3%)
**CT**	
Vertebral fractures (patients)	22 (2.2%)
Number of vertebral fractures	32
SBAC-L1(HU)	194.0 (±69.0)
SBAC-L1 ≤ 145 HU	163 (16.1%)
SBAC-L1 >145 HU	850 (83.9%)
**DXA**	
*Femoral neck*	
BMD (g/cm²)	1.098 (±0.15)
T-score (SD)	0.9 (±1.5)
Osteoporosis	2 (0.2%)
Osteopenia	65 (6.4%)
*Hip*	
BMD (g/cm²)	1.169 (±0.15)
T-score (SD)	1.4 (±1.6)
Osteoporosis	0 (0%)
Osteopenia	21 (2.1%)
*Spine* (*L1-L4*)	
BMD (g/cm²)	1.273 (±0.16)
T-score (SD)	0.8 (±1.7)
Osteoporosis	8 (0.79%)
Osteopenia	73 (7.2%)
*Osteoporosis* on at least one site	8 (0.8%)
*Osteopenia* on at least one site	119 (11.7%)
Fat mass (kg)	60.5 (±17.8)
Lean mass (kg)	55.6 (±14.2)
Fat mass/total weight (%)	51.0% (±6.3)
Android fat mass/Gynoid fat mass	1.6 (±0.7)
Lipodystrophy index	0.8 (±0.3)
Sarcopenia	10.1 (±1.9)
VAT (g)	2407.0 (±1594.0)

Data are presented as n (%) for dichotomous variables, mean (SD) for continuous demographic variables with normal distribution and median (interquartile range) with the nonnormal distribution. BMI: body mass index; CT: computed tomography; SBAC-L1: Scanographic bone attenuation coefficient of L1; HU: Hounsfield unit; DXA: dual-energy X-ray absorptiometry; BMD: bone mineral density; SD: standard deviation; VAT: visceral adipose tissue. The percentage was calculated based on the available data for each variable. Osteoporosis was defined by a T-score ≤ −2.5 SD at any measured location, and osteopenia was defined by −2.5 SD < T-score ≤ −1 SD. Vitamin D deficiency was defined by a level below 30 ng/mL.

**Table 2 jcm-11-06114-t002:** Characteristics and comparison of patients with both DXA and CT (<2 years between the 2 exams) according to SBAC-L1 status (n = 1013).

		SBAC-L1		Univariate			Multivariate		
	n	>145 HU	≤145 HU	*p* Value	OR	CI 95	*p* Value	OR	CI 95
**Demographic data**		850 patients	163 patients						
Age (year)	1013	47.0 (±16.0)	59.0 (±12.0)	**0.0001**	**1.1**	**[1.10–1.14]**	**0.0001**	**1.1**	**[1.08–1.16]**
Sex (women)	1013	678 (79.8%)	98 (60.1%)	**0.0001**	**0.4**	**[0.27–0.55]**	0.6	1.3	[0.47–3.82]
BMI (g/cm²)	1013	44.8 (±8.5)	43.0 (±7.4)	**0.01**	**0.9**	**[0.9–0.99]**	**0.0001**	**0.9**	**[0.82–0.92]**
Diabetes	1013	245 (28.8%)	71 (43.6%)	**0.0001**	**1.9**	**[1.35–2.68]**	**0.3**	**0.8**	**[0.43–1.32]**
Cardiovascular risk factor	1013	460 (54.1%)	124 (76.1%)	**0.0001**	**2.7**	**[1.84–3.96]**	0.8	1.1	[0.61–1.94]
Tabacco Smokers Never smokers	981	337 (39.6%) 487 (57.3%)	75 (46.0%) 82 (50.3%)	0.11	1.3	[0.94–1.86]			
Alcohol	981	31 (3.6%)	11 (6.7%)	0.07	1.9	[0.95–3.92]			
Vitamin D deficiency	922	708 (83.3%)	134 (82.2%)	0.5	0.8	[0.45–1.48]			
**CT**	1013	850 patients	163 patients						
Vertebral fractures (patients)		13 (1.5%)	9 (5.5%)	**0.003**	**3.8**	**[1.58–8.96]**			
**DXA**	1013	850 patients	163 patients						
Femoral neck									
BMD (g/cm²)		1.120 (±0.14)	0.985 (±0.16)	**0.0001**	**0.001**	**[0.0001–0.004]**	**0.03**	**0.04**	**[0.005–0.33]**
T-score (SD)		1.1 (±1.3)	−0.2 (±1.6)	**0.0001**	**0.3**	**[0.28–0.42]**			
Osteoporosis		2 (0.2%)	0 (0%)						
Osteopenia		25 (2.9%)	40 (24.5%)	**0.0001**	**11.1**	**[6.45–18.97]**			
*Hip*									
BMD (g/cm²)		1.189 (±0.14)	1.067 (±0.14)	**0.0001**	**0.002**	**[0.001–0.009]**			
T-score (SD)		1.5 (±1.4)	0.3 (±1.3)	**0.0001**	**0.4**	**[0.30–0.45]**			
Osteoporosis		0 (0%)	0 (0%)						
Osteopenia		10 (1.2%)	11 (6.7%)	**0.0001**	**6.0**	**[2.51–14.48]**			
*Spine (L1-L4)*									
BMD (g/cm²)		1.295 (±0.15)	1.159 (±0.14)	**0.0001**	**0.001**	**[0.0001–0.006]**	**0.0001**	**0.001**	**[0.0001–0.008]**
T-score (SD)		1.0 (±1.5)	−0.2 (±1.3)	**0.0001**	**0.4**	**[0.38–0.53]**			
Osteoporosis		3 (0.4%)	5 (3.1%)	**0.003**	**9.0**	**[2.12–38.02]**			
Osteopenia		39 (4.6%)	34 (20.9%)	**0.0001**	**5.6**	**[3.41–9.30]**			
Osteoporosis on at least on site		3 (0.4%)	5 (3.1%)	**0.003**	**8.9**	**[2.11–37.68]**			
Osteopenia on at least one site		60 (7.1%)	59 (36.2%)	**0.0001**	**7.8**	**[5.12–11.91]**			
Fat mass (kg)		60.9 (±17.8)	59.3 (±19.3)	0.09	0.99	[0.98–1.00]			
Lean mass (kg)		55.4 (±13.4)	57.9 (±20.2)	**0.007**	**1.02**	**[1.01–1.04]**	**0.002**	**1.1**	**[1.03–1.13]**
Fat mass/total weight (%)		51.2% (±6.2)	49.4% (±7.9)	**0.0001**	**0.94**	**[0.92–0.97]**			
Android fat mass/Gynoid fat mass		1.6 (±0.7)	1.8 (±0.8)	**0.001**	**2.3**	**[1.69–3.08]**			
Lipodystrophy index		0.8 (±0.3)	0.6 (±0.3)	**0.0001**	**0.068**	**[0.026–0.180]**	0.9	1.1	[0.18–7.47]
Sarcopenia		10.1 (±1.9)	10.4 (±2.1)	0.39	1.05	[0.94–1.18]			
VAT (g)		2301.5 (±1472.8)	3104.0 (±2372.0)	**0.0001**	**1**	**[1–1.001]**	0.02	1	[1.0–1.001]

BMI: body mass index; GBP: gastric bypass; SADI: single anastomosis duodeno-ileal bypass; CT computed tomography; SBAC-L1: Scanographic bone attenuation coefficient of L1; HU: Hounsfield unit; DXA: dual-energy X-ray absorptiometry; BMD: bone mineral density; SD: standard deviation; VAT: visceral adipose tissue. The percentage was calculated based on the available data for each variable. The results in bold are statistically significant (*p* < 0.05). For the multivariate analysis, all significant variables in the univariate analysis were included. For the bone study on DXA, BMD and T-score were interdependent; therefore, for univariate analysis, we decided to include only the BMD for each studied location. Data are presented as n (%) for dichotomous variables, mean (SD) for continuous demographic variables with normal distribution and median [interquartile range] with the nonnormal distribution. Osteoporosis was defined by a T-score ≤ −2.5 SD at any measured location, and osteopenia was defined by −2.5 SD < T-score ≤ −1 SD. Vitamin D deficiency was defined by a level below 30 ng/mL. Only significant variables from the univariate analysis were included in the multivariate analysis; variables with missing data were not included (like VF, smoking status, alcohol et vitamin D); osteopenia, osteoporosis and T-score were not included because they are associated with BMD; ratio variables were also excluded.

## Data Availability

The data presented in this study are available on request from the corresponding author.

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
