# Peer review of "Prevalence of Osteoporosis Assessed by DXA and/or CT in Severe Obese Patients"

_jcm, 2022, doi:10.3390/jcm11206114_

Round 1
Reviewer 1 Report
General comments
The study is confusing as many different populations are investigated in the same study (Figure 1). In my view you should limit the study to the population that have been DEXA scanned and CT scanned i.e. 1013 patients. This will make the manuscript more focused and easier to follow. This is still a large study population.
Specific comments
The title is a bit strange. It is not until the last paragraph of the introduction that the term “bone frailty” is defined, and at least for DEXA the definition is corresponds to the WHO definition of osteoporosis. What was the rationale behind the title? Why not: “Prevalence of osteoporosis assessed by DEXA and/or CT in severe obese patients”?
Page 1, line 45
Osteoporosis is defined in [5] as: “Osteoporosis. A value for BMD or BMC 2.5 SD or more below the young adult mean”. Please correct.
Page 2, line 63 and throughout the manuscript
Please do not referrer to the threshold as a “fracture threshold”.
In you data there are 15 patients out of 978 patients with a fracture (= 1.53%) with a SBAC-L1 > 145 HU and 12 patients out of 201 patients with a fracture (= 5.97%) with a SBAC-L1 £ 145 HU. Thus, it is erroneous to referrer to the threshold as a fracture threshold since this is not supported by you own data. A “fracture threshold” indicate that having a SBAC-L1 below the limit means that the patient has a fracture, and that has only 6% of your patients!
Page 2, lines 64-66
In addition, area BMD (aBMD) is dependent on the size (thickness) of the bone and is therefore not a real density. In contrast, the volumetric BMD assessed by QCT is a real density and thus independent of bone size.
Page 3, lines 97-111
Please state the voxel size, the X-ray tube voltage and current, and the integration time used in the CT-scans.
How many layers of voxels were included in the analyzed area? Only one since you denote it ROI (region of interest) rather than VOI (volume of interest)? Thus, please state the thickness of the analyzed area.
“lector”? do you mean “investigator”?
Page 3, line 131
Were the ages normally distributed? If not, you cannot use a Student’s t test.
Page 5, line 172 and page 6 line 186
What does the “o” mean?
Page 5, line 173 and throughout the manuscript
The standard deviation also has a unit, thus “195.1 HU (±52.7)” -> “195.1 (± 52.7) HU”
Page 8, Figure 2
Why is it the femoral neck T-scores and not the T-scores of the spine that are plotted for the patients with vertebral fractures?
The text on the axis are difficult to read, especially in Figure 2C. What does CASL-1_UH mean and what unit does it have?
Page 11, lines 330-335
You cannot underestimate the risk of vertebral fractures, when you have not estimated the risk of fracture with DEXA, just identified the patients, which had osteoporosis according to the WHO criteria for osteoporosis.
Tables
It is highly confusing to use ( ) for both percentage and SD. Moreover the “%” is not a unit, but a part of the number meaning ×0.01. Consequently, 15 out of 978 should be written as 1.53% and not as 1.53. Please use the normal convention of “mean ± SD” instead.
Author Response
You can not see the figures' modification in these notes. Please look at the uploaded file.
Point 1: General comments
The study is confusing as many different populations are investigated in the same study (Figure 1). In my view you should limit the study to the population that have been DEXA scanned and CT scanned i.e. 1013 patients. This will make the manuscript more focused and easier to follow. This is still a large study population.
Specific comments
Response 1: We have initially preferred to present all the data to show that there is not significant difference between patients with DXA and CT and patients who performed only one of this exam (Table 1). We removed data about patients with only DXA or CT to be clearer.
Point 2: The title is a bit strange. It is not until the last paragraph of the introduction that the term “bone frailty” is defined, and at least for DEXA the definition is corresponds to the WHO definition of osteoporosis. What was the rationale behind the title? Why not: “Prevalence of osteoporosis assessed by DEXA and/or CT in severe obese patients”?
Response 2: Thank you for this remark. We have changed the title with “Prevalence of osteoporosis assessed by DXA and/or CT in severe obese patients”.
We have previously mentioned “bone fragility” and not osteoporosis in the title because the SBAC-L1 threshold was not an official threshold to diagnose osteoporosis.
Point 3: Page 1, line 45. Osteoporosis is defined in [5] as: “Osteoporosis. A value for BMD or BMC 2.5 SD or more below the young adult mean”. Please correct.
Response 3: we have added this data in the sentence.
Point 4: Page 2, line 63 and throughout the manuscript
Please do not referrer to the threshold as a “fracture threshold”.
In your data there are 15 patients out of 978 patients with a fracture (= 1.53%) with a SBAC-L1 > 145 HU and 12 patients out of 201 patients with a fracture (= 5.97%) with a SBAC-L1 £ 145 HU. Thus, it is erroneous to referrer to the threshold as a fracture threshold since this is not supported by you own data. A “fracture threshold” indicate that having a SBAC-L1 below the limit means that the patient has a fracture, and that has only 6% of your patients!
Response 4: thank you for this remark. You are wright. We changed all “fracture threshold” to “threshold”.
Point 5: Page 2, lines 64-66
In addition, area BMD (aBMD) is dependent on the size (thickness) of the bone and is therefore not a real density. In contrast, the volumetric BMD assessed by QCT is a real density and thus independent of bone size.
Response 5: You are wright, thank you for this remark but in this study, we did not use QCT but “standard” CT to evaluate bone mineral density, through the SBAC-L1.
Point 6: Page 3, lines 97-111
Please state the voxel size, the X-ray tube voltage and current, and the integration time used in the CT-scans.
Response 6: Sadly, some of these data are missing. Concerning voltage, most of the CT are performed with a 120 kV voltage (min 100 kV – max 140 kV, adapted to fat mass quantity). The mean slice thickness was 1.80 (±0.91) mm.
Point 7: How many layers of voxels were included in the analyzed area? Only one since you denote it ROI (region of interest) rather than VOI (volume of interest)? Thus, please state the thickness of the analyzed area.
Response 7: A ROI was drawn on CT axial slice but the measure (mean attenuation coefficient in HU) takes into account the thickness of this slice. The mean thickness was 1.87 (±0.56).
Point 8: “lector”? do you mean “investigator”?
Response 8: you are wright, we have changed “lector” to investigator.
Point 9: Page 3, line 131
Were the ages normally distributed? If not, you cannot use a Student’s t test.
Response 9: Ages were effectively normally distributed; the normality was assessed by Kolmogorov–Smirnov test. For other quantitative variables, without normality distribution, Mann–Whitney U test was used.
Point 10: Page 5, line 172 and page 6 line 186
What does the “o” mean?
Response 10: “o” is a tabulation mark. We changed its appearance.
Point 11: Page 5, line 173 and throughout the manuscript
The standard deviation also has a unit, thus “195.1 HU (±52.7)” -> “195.1 (± 52.7) HU”
Response 11: Thank you for this remark. We modified all results in abstract, results and discussion parts.
Point 12: Page 8, Figure 2
Why is it the femoral neck T-scores and not the T-scores of the spine that are plotted for the patients with vertebral fractures?
Response 12: we have used femoral neck T-score because in obese patients, spine T-score is often overestimated, due to osteoarthritis, worsened by overweight. We also known that the femoral neck T-score is better correlated with the fracture risk, especially for vertebral fracture risk in the literature.
Since the two reviewers have formulated the same remark, we proposed to change this figure, to include spine T-score instead of femoral neck T-score.
Point 13: The text on the axis are difficult to read, especially in Figure 2C. What does CASL-1_UH mean and what unit does it have?
Response 13: Thank you for this remark to improve our article quality. We have made some modifications.
Point 14: Page 11, lines 330-335
You cannot underestimate the risk of vertebral fractures, when you have not estimated the risk of fracture with DEXA, just identified the patients, which had osteoporosis according to the WHO criteria for osteoporosis.
Response 14: Thank you for this remark, you are wright but we also know that patients with osteoporosis are with a high fracture risk, higher than patients without osteoporosis on DXA. But in this study, we highlighted that only few patients are screened as osteoporosis with the DXA.
Point 15: Tables
It is highly confusing to use ( ) for both percentage and SD. Moreover the “%” is not a unit, but a part of the number meaning ×0.01. Consequently, 15 out of 978 should be written as 1.53% and not as 1.53. Please use the normal convention of “mean ± SD” instead.
Response 15: In all tables, we added % and ± for SD, to reduce the confusion.

Reviewer 2 Report
Reviewer Comments-Journal of Clinical Medicine
Prevalence of bone fragility on DXA and CT in Patients with severe obesity
This study investigated the respective measures of bone fragility (t-scores and SBAC-L1) by DXA and CT in patients with severe obesity. In patients that received both a DXA and CT 22 patients had a VF; 44% of these patients would have been identified with a SBAC< 145HU by CT, and 37% as osteoporotic or osteopenic by DXA t-scores. A SBAC-L1 below the fracture threshold was associated with increased age and positive diabetes or cardiovascular risk status, while being female or having a lower BMI decreased the odds of having a SBAC-L1 < 145 HU. However, in a multivariate analysis the odds ratio of all variables trended towards OR=1.
General comments:
While this paper may contain novel data as the first to look SBAC-L1 in patients with severe obesity, the presentation of the results was convoluted and hindered the reader’s interpretation. The tabular presentation of the data and figures could be much clearer and much of the confusion was tied to vague statistical methods. In the specific comments below are some examples illuminating what was unclear and suggestions to guide the authors on potential revisions.
The discussion was underdeveloped and lacked insightful interpretation of the data. The discussion should be more than a retelling of the results. Besides comparing prevalence rates to literature, there were only small bits of information that was already mentioned in the introduction. The authors may consider adding more detail regarding how their data fits within the context of the conflicting themes in literature (low BMI vs high BMI and fracture risk), how this may inform new practices DXA vs CT in these populations, or how the risk factors evaluated in this study may be more important in obese vs non-obese populations. These are all merely suggestions, but the bottom line is that the discussion needs to be elevated with thoughtful interpretations from the authors.
The phrase bone fragility is used as a general catch all phrase and also as a specific “measure” associated with a multitude of real measurements (e.g. SBAC-L1, t-scores, fracture status or BMD). The paper would benefit from using more intentional wordage throughout but at the very least when discussing specific results please be more explicit about the actual measurement of interest. For example, line 237-238 is an instance where this might not be clear. Additionally, I would suggest changing the title (and many similar phrases throughout the paper) of “the prevalence of bone fragility on DXA and CT…” to something more meaningful such as “Measurements of bone fragility using DXA and CT…”.
Specific Comments:
1. Define grade 2 and 3 obesity. In the intro obesity is defined as mild, severe, and morbid. Is it the same? If so please say so explicitly since “grade 2 and 3” is used throughout the rest of the study
2. Of the 1013 patients with CT and DXA exams within 2 years, what was the average time between exams?
3. How was missing data dealt with in the statistical analysis? (For example, smoking status or vitamin D was not reported for all 1386 patients)
4. Please define TAV.
5. It may be more appropriate to reference the tables and figures within the text and not in the section headers.
6. One of the most interesting findings here is that high fat mass has been associated with many factors that could affect bone health and fragility, but the current results don’t show a strong influence of BMI on fracture risk (SBAC-L1 outcome). This may be a point to emphasize in the discussion.
7. This population may have been a bit young to expect a detectable level of bone degradation that would be classified by DXA as osteoporosis or osteopenia. With this limitation in mind, it makes a stronger case for the use of CT and the SBAC-L1 evaluation for early bone fragility and fracture risk, a point the authors may want to expand on in the discussion.
8. Would getting a CT scan be feasible? What is the practicality of suggesting using CT as a screening tool in this population?
9. Was there a relationship between t-scores and SBAC-L1 values in the patients that had both DXA and CT? And what would a such a relationship (or lack of) tell us about the two common ways of assessing bone fragility in obese population?
10. Line 61: These references seem inappropriately placed as they primarily report on obesity, bariatric surgery, risk factors of factor etc. in osteoporotic populations. While relevant and important references to support other parts of this intro the reference placed here should be supporting the previous statement on using CT to evaluate VFs with SBAC-L1 measures.
11. Line 95-96: Please clarify what is meant by “BMD measurement was coupled with analysis of the distribution of fat and lean mass.”
12. Line 226: By protective factors do you mean the odds are less likely of have VF or SBAC-L1<145HU? And is it appropriate to call high BMI protective when it is nearly no effect on the odds of the outcome (i.e. very close to OR=1). See similar phrasing on lines 237-238.
13. In table 2 what does the significance value mean? Significantly different between >145HU and <145HU groups or significant with respect to the individual test for the different variables? Like the sex is different between the groups or it was significant when added as variable in the logistics regression analysis? Maybe this should be addressed in the table caption.
14. Which variables were included in the multivariate analysis? Logistics regression could be presented as table the combination pf variables included in the models to predict the SBAC-L1>145 or <145 HU groups and the p-value associated with the various combination. (I see know in the figure caption you have mentioned only the variables significant in univariate analysis and those that were not interdependent were included. This information should be in the main text. Also, is the choice to only include significant variables from the univariate analysis in the multivariate model appropriate? This could be influenced by the presence of missing data in the variables left out- smoking status, alcohol and vit D- and may miss out on interaction effects. The current choice may be appropriate but please justify it in the discussion and note the limitations of leaving out other data.)
15. Quality of graphs in Figure 2 is poor. Stretched and too small of a font to read. Also the axis labels in Figure 2C are not readable or informative titles.
16. It may be more comparable to show spine t-scores in Figure 2B (instead of femoral neck t-scores) since Figure 2A is SBAC measure of the lumbar.

Author Response
You can not see the figures' modification in these notes. Please look at the uploaded file.
This study investigated the respective measures of bone fragility (t-scores and SBAC-L1) by DXA and CT in patients with severe obesity. In patients that received both a DXA and CT 22 patients had a VF; 44% of these patients would have been identified with a SBAC< 145HU by CT, and 37% as osteoporotic or osteopenic by DXA t-scores. A SBAC-L1 below the fracture threshold was associated with increased age and positive diabetes or cardiovascular risk status, while being female or having a lower BMI decreased the odds of having a SBAC-L1 < 145 HU. However, in a multivariate analysis the odds ratio of all variables trended towards OR=1.
Point 1: General comments:
While this paper may contain novel data as the first to look SBAC-L1 in patients with severe obesity, the presentation of the results was convoluted and hindered the reader’s interpretation. The tabular presentation of the data and figures could be much clearer and much of the confusion was tied to vague statistical methods. In the specific comments below are some examples illuminating what was unclear and suggestions to guide the authors on potential revisions.
Point 2: The discussion was underdeveloped and lacked insightful interpretation of the data. The discussion should be more than a retelling of the results. Besides comparing prevalence rates to literature, there were only small bits of information that was already mentioned in the introduction. The authors may consider adding more detail regarding how their data fits within the context of the conflicting themes in literature (low BMI vs high BMI and fracture risk), how this may inform new practices DXA vs CT in these populations, or how the risk factors evaluated in this study may be more important in obese vs non-obese populations. These are all merely suggestions, but the bottom line is that the discussion needs to be elevated with thoughtful interpretations from the authors.
Responses 1 and 2: Thank you for your remarks to improve the quality of our manuscript. We answered to your questions and make some modifications in the article. We also focused the article on the results about the populations with both DXA and CT (n=1013) to be clearer.
Point 3: The phrase bone fragility is used as a general catch all phrase and also as a specific “measure” associated with a multitude of real measurements (e.g. SBAC-L1, t-scores, fracture status or BMD). The paper would benefit from using more intentional wordage throughout but at the very least when discussing specific results please be more explicit about the actual measurement of interest. For example, line 237-238 is an instance where this might not be clear.
Response 3: Bone fragility was described in the objective, according to the exam. For DXA, bone fragility was defined by a T-score ≤ -2.5 SD and for CT, by a SBAC-L1 ≤ 145HU. We changed the phrase “bone fragility” to be more precise in different places in the article.
Point 4: Additionally, I would suggest changing the title (and many similar phrases throughout the paper) of “the prevalence of bone fragility on DXA and CT…” to something more meaningful such as “Measurements of bone fragility using DXA and CT…”.
Response 4: we have changed the title for “Prevalence of osteoporosis assessed by DXA and/or CT in severe obese patients”, according to the remark from another reviewer.
Specific Comments:
Point 5: Define grade 2 and 3 obesity. In the intro obesity is defined as mild, severe, and morbid. Is it the same? If so please say so explicitly since “grade 2 and 3” is used throughout the rest of the study
Response 5: Grade 2 correspond to a BMI between 35 and 40 kg/m2 and severe obesity; Grade 3 correspond to BMI over 40 kg/m2 and morbid obesity. Grades 2 and 3 correspond to the patients eligible for bariatric surgery. We have clarified that in the introduction part.
Point 6: Of the 1013 patients with CT and DXA exams within 2 years, what was the average time between exams?
Response 6: Of the 1013 patients with both DXA and CT within 2 years, the average time between the exams was 275.4 (±174.4) days. We added this data in the results part.
Point 7: How was missing data dealt with in the statistical analysis? (For example, smoking status or vitamin D was not reported for all 1386 patients)
Response 7: We have analyzed only the available data. The percentage of missing data is low and quite similar between the populations, so we can consider that missing data have low impact on bivariate analysis. The presence of such missing data can not conduct to an interaction effect in the logistic regression model.
Point 8: Please define TAV.
Response 8: Thank you for this remark. We have forgotten to define TAV in the article. Sorry. We corrected that and changed for VAT: visceral adipose tissue
Point 9: It may be more appropriate to reference the tables and figures within the text and not in the section headers.
Response 9: Thank you. We have change that to improve the presentation.
Point 10: One of the most interesting findings here is that high fat mass has been associated with many factors that could affect bone health and fragility, but the current results don’t show a strong influence of BMI on fracture risk (SBAC-L1 outcome). This may be a point to emphasize in the discussion.
Response 10: Thank you for this remark, we added it in the discussion.
Point 11: This population may have been a bit young to expect a detectable level of bone degradation that would be classified by DXA as osteoporosis or osteopenia. With this limitation in mind, it makes a stronger case for the use of CT and the SBAC-L1 evaluation for early bone fragility and fracture risk, a point the authors may want to expand on in the discussion.
Response 11: Yes, you are wright. Thank you for this interestingly remark. We added it in the discussion.
Point 12: Would getting a CT scan be feasible? What is the practicality of suggesting using CT as a screening tool in this population?
Response 12: In our opinion, a CT is not feasible just to explore bone fragility, unless there is vertebral fracture suspicion. This exam is too expansive and expose the patient to too much radiation compared to DXA. In the future, new low dose spine CT could offer a new approach to detect vertebral fracture and to measure SBAC-L1. In this case, information about thorax and pelvic abdominal regions would be missing. However, CT are very frequently performed to other indication, so we can use them opportunistically to screen patients at risk of bone fragility.
We added this consideration in the conclusion.
Point 13: Was there a relationship between t-scores and SBAC-L1 values in the patients that had both DXA and CT? And what would a such a relationship (or lack of) tell us about the two common ways of assessing bone fragility in obese population?
Response 13: Thank you for this interestingly question.
The correlation coefficient was 0.48 (CI95=0.43-0.53) between femoral neck T-score and SBAC-L1 and 0.43 (CI95=0.36-0.49) between femoral neck BMD and SBAC-L1; 0.46 (CI95=0.40-0.51) between spine T-score and SBAC-L1 and 0.42 (CI95=0.37-0.48) between spine BMD and SBAC-L1 (Pearson coefficient was used for BMD and Spearman’s rho for T-score, according to the variables distribution). So, the correlation was positive but poor.
This lack of relationship could be explained by the difference in term of measure, between the 2 exams. CT excluded cortical bone and is less influenced by the body fat mass and is more representative of the real bone mass, than DXA. Moreover, one advantage of CT compared to DXA is its ability to accurately identify unsuspected osteoporotic vertebral fractures, which clearly diagnoses osteoporosis independent of the patient’s DXA T-score.
We added this data in the article (results and discussion parts).
Point 14: Line 61: These references seem inappropriately placed as they primarily report on obesity, bariatric surgery, risk factors of factor etc. in osteoporotic populations. While relevant and important references to support other parts of this intro the reference placed here should be supporting the previous statement on using CT to evaluate VFs with SBAC-L1 measures.
Response 14: thank you for this remark, you have wright. It is an error and we corrected it with the appropriate references.
Point 15: Line 95-96: Please clarify what is meant by “BMD measurement was coupled with analysis of the distribution of fat and lean mass.”
Response 15: The DXA allowed to measure simultaneously BMD at any site and lean and fat mass repartition on the body, on the same exam.
Point 16: Line 226: By protective factors do you mean the odds are less likely of have VF or SBAC-L1<145HU? And is it appropriate to call high BMI protective when it is nearly no effect on the odds of the outcome (i.e. very close to OR=1).
Point 17: See similar phrasing on lines 237-238.
Responses 16 and 17: By protective factor, we mean the odds are less likely of have SBAC-L1 ≤ 145HU (the precision was added in the article). The relation with VF cannot be evaluated, due to the low number of fractures in this population. For BMI, you are wright, OR is verry close to 1 but it stays statistically significant.
Point 18: In table 2 what does the significance value mean? Significantly different between >145HU and <145HU groups or significant with respect to the individual test for the different variables? Like the sex is different between the groups or it was significant when added as variable in the logistics regression analysis? Maybe this should be addressed in the table caption.
Response 18: In the table 2, the p value was evaluated between the variables in SBAC-L1 > 145HU and SBAC-L1 ≤ 145 HU groups. Sex is significantly different in the 2 groups: there is 78.5% of women in >145 HU group and 59.8% in ≤ 145 HU group. We have precise that in the table caption: “The p-value was performed to compare the data between the 2 groups: SBAC-L1 > 145 HU and SBAC-L1 ≤ 145 HU”.
Point 19: Which variables were included in the multivariate analysis? Logistics regression could be presented as table the combination pf variables included in the models to predict the SBAC-L1>145 or <145 HU groups and the p-value associated with the various combination. (I see know in the figure caption you have mentioned only the variables significant in univariate analysis and those that were not interdependent were included. This information should be in the main text. Also, is the choice to only include significant variables from the univariate analysis in the multivariate model appropriate? This could be influenced by the presence of missing data in the variables left out- smoking status, alcohol and vit D- and may miss out on interaction effects. The current choice may be appropriate but please justify it in the discussion and note the limitations of leaving out other data.)
Response 19: Thank you for this remark.
In the multivariate analysis, we included only significant variables from the univariate analysis, variables with missing data were not included (like VF, smoking status, alcohol et vitamin D). Osteopenia, osteoporosis and T-score were not included because they are associated to BMD. Ratio variables were also excluded.
We added the information about the included variables for multivariate analysis in the main text and in the table 2 legend.
Point 20: Quality of graphs in Figure 2 is poor. Stretched and too small of a font to read. Also the axis labels in Figure 2C are not readable or informative titles.
Response 20: Thank you for this remark to improve our article quality. We have made some modifications.
Point 21: It may be more comparable to show spine t-scores in Figure 2B (instead of femoral neck t-scores) since Figure 2A is SBAC measure of the lumbar.
Response 21: we have used femoral neck T-score because in obese patients, spine T-score is often overestimated, due to osteoarthritis, worsened by overweight. Since the two reviewers have formulated the same remark, we proposed to change this figure, to include spine T-score instead of femoral neck T-score.

Round 2
Reviewer 1 Report
The manuscript is much improved. I have a few issues listed below.
Answers to my previous comments
Please, provide data about voxel size in the manuscript and not only in the response to the reviewer. It is unfortunate that key scanning data has been lost. Are these data not embedded as metadata in your image files?
Similar for the thickness of the bone slice investigated: This information must be included in the manuscript.
Page 12, lines 307-309 & page 13, lines 354-356
You are not estimating mass but density with the two techniques. Secondly, it is not the technique as such that is representative of the real bone density but it is the values obtained with the technique that is more representative of the real bone mass. “CT excluded cortical bone and is less influenced by the body fat mass and is more representative of the real bone mass, than DXA.” ->
“CT excluded cortical bone and is less influenced by the body fat and the values obtained are more representative of the real bone density than those obtained by DXA”
Page 13, line 321
“This exam is too expansive and expose…” did you mean “This examination is expensive and expose…” ?
Author Response
Point 1:
The manuscript is much improved. I have a few issues listed below.
Answers to my previous comments
Please, provide data about voxel size in the manuscript and not only in the response to the reviewer. It is unfortunate that key scanning data has been lost. Are these data not embedded as metadata in your image files?
Similar for the thickness of the bone slice investigated: This information must be included in the manuscript.
Response 1: Sadly, data on the voxel size (who could be calculated with the matrix and acquisition volume) were not available in this study. We take into account your remark for further studies.
We added a sentence about voxel size in the manuscript (results part).
Point 2: Page 12, lines 307-309 & page 13, lines 354-356
You are not estimating mass but density with the two techniques. Secondly, it is not the technique as such that is representative of the real bone density but it is the values obtained with the technique that is more representative of the real bone mass. “CT excluded cortical bone and is less influenced by the body fat mass and is more representative of the real bone mass, than DXA.” ->
“CT excluded cortical bone and is less influenced by the body fat and the values obtained are more representative of the real bone density than those obtained by DXA”
Response 2: Thank you for this remark. We changed the sentence.
Point 3: Page 13, line 321
“This exam is too expansive and expose…” did you mean “This examination is expensive and expose…” ?
Response 3 : You are wright. We modified the sentence.
